# DC3: A LEARNING METHOD FOR OPTIMIZATION WITH HARD CONSTRAINTS

**Priya L. Donti[1,*], David Rolnick[2,*], J. Zico Kolter[1,3]**
[1]Carnegie Mellon University, [2]McGill University and Mila, [3]Bosch Center for AI
`pdonti@cs.cmu.edu, drolnick@cs.mcgill.ca, zkolter@cs.cmu.edu`

## ABSTRACT

Large optimization problems with hard constraints arise in many settings, yet classical solvers are often prohibitively slow, motivating the use of deep networks as cheap "approximate solvers." Unfortunately, naive deep learning approaches typically cannot enforce the hard constraints of such problems, leading to infeasible solutions. In this work, we present Deep Constraint Completion and Correction (DC3), an algorithm to address this challenge. Specifically, this method enforces feasibility via a differentiable procedure, which implicitly completes partial solutions to satisfy equality constraints and unrolls gradient-based corrections to satisfy inequality constraints. We demonstrate the effectiveness of DC3 in both synthetic optimization tasks and the real-world setting of AC optimal power flow, where hard constraints encode the physics of the electrical grid. In both cases, DC3 achieves near-optimal objective values while preserving feasibility.

## 1 INTRODUCTION

Traditional approaches to constrained optimization are often expensive to run for large problems, necessitating the use of function approximators. Neural networks are highly expressive and fast to run, making them ideal as function approximators. However, while deep learning has proven its power for unconstrained problem settings, it has struggled to perform well in domains where it is necessary to satisfy hard constraints at test time. For example, in power systems, weather and climate models, materials science, and many other areas, data follows well-known physical laws, and violation of these laws can lead to answers that are unhelpful or even nonsensical. There is thus a need for fast neural network approximators that can operate in settings where traditional optimizers are slow (such as non-convex optimization), yet where strict feasibility criteria must be satisfied.

In this work, we introduce Deep Constraint Completion and Correction (DC3), a framework for applying deep learning to optimization problems with hard constraints. Our approach embeds differentiable operations into the training of the neural network to ensure feasibility. Specifically, the network outputs a partial set of variables with codimension equal to the number of equality constraints, and "completes" this partial set into a full solution. This completion process guarantees feasibility with respect to the equality constraints and is differentiable (either explicitly, or via the implicit function theorem). We then fix any violations of the inequality constraints via a differentiable correction procedure based on gradient descent. Together, this process of completion and correction enables feasibility with respect to all constraints. Further, this process is fully differentiable and can be incorporated into standard deep learning methods.

Our key contributions are:

- **Framework for incorporating hard constraints.** We describe a general framework, DC3, for incorporating (potentially non-convex) equality and inequality constraints into deep-learning-based optimization algorithms.

- **Practical demonstration of feasibility.** We implement the DC3 algorithm in both convex and non-convex optimization settings. We demonstrate the success of the algorithm in producing approximate solutions with significantly better feasibility than other deep learning approaches, while maintaining near-optimality of the solution.

---

[*]These authors contributed equally.

- **AC optimal power flow.** We show how the general DC3 framework can be used to optimize power flows on the electrical grid. This difficult non-convex optimization task must be solved at scale and is especially critical for renewable energy adoption. Our results greatly improve upon the performance of general-purpose deep learning methods on this task.

## 2 RELATED WORK

Our approach is situated within the broader literature on fast optimization methods, and draws inspiration from literature on implicit layers and on incorporating constraints into neural networks. We briefly describe each of these areas and their relationship to the present work.

**Fast optimization methods.** Many classical optimization methods have been proposed to improve the practical efficiency of solving optimization problems. These include general techniques such as constraint and variable elimination (i.e., the removal of non-active constraints or redundant variables, respectively), as well as problem-specific techniques (e.g., KKT factorization techniques in the case of convex quadratic programs) (Nocedal & Wright, 2006). Our present work builds upon aspects of this literature, applying concepts from variable elimination to reduce the number of degrees of freedom associated with the optimization problems we wish to solve.

In addition to the classical optimization literature, there has been a large body of literature in deep learning that has sought to approximate or speed up optimization models. As described in reviews on topics such as combinatorial optimization (Bengio et al., 2020) and optimal power flow (Hasan et al., 2020), ML methods to speed up optimization models have thus far taken two main approaches. The first class of approaches, akin to work on surrogate modeling (Koziel & Leifsson, 2013), has involved training machine learning models to map directly from optimization inputs to full solutions. However, such approaches have often struggled to produce solutions that are both feasible and (near-)optimal. The second class of approaches has instead focused on employing machine learning approaches alongside or in the loop of optimization models, e.g., to learn warm-start points (see, e.g., Baker (2019) and Dong et al. (2020)) or to enable constraint elimination techniques by predicting active constraints (see, e.g., Misra et al. (2018)). We view our work as part of the former set of approaches, but drawing important inspiration from the latter: that employing structural knowledge about the optimization model is paramount to achieving both feasibility and optimality.

**Constraints in neural networks.** While deep learning is often thought of as wholly unconstrained, in reality, it is quite common to incorporate (simple) constraints within deep learning procedures. For instance, softmax layers encode simplex constraints, sigmoids instantiate upper and lower bounds, ReLUs encode projections onto the positive orthant, and convolutional layers enforce translational equivariance (an idea taken further in general group-equivariant networks (Cohen & Welling, 2016)). Recent work has also focused on embedding specialized kinds of constraints into neural networks, such as conservation of energy (see, e.g., Greydanus et al. (2019) and Beucler et al. (2019)), and homogeneous linear inequality constraints (Frerix et al., 2020). However, while these represent common "special cases," there has to date been little work on building more general hard constraints into deep learning models.

**Implicit layers.** In recent years, there has been a great deal of interest in creating structured neural network layers that define implicit relationships between their inputs and outputs. For instance, such layers have been created for SAT solving (Wang et al., 2019), ordinary differential equations (Chen et al., 2018), normal and extensive-form games (Ling et al., 2018), rigid-body physics (de Avila Belbute-Peres et al., 2018), sequence modeling (Bai et al., 2019), and various classes of optimization problems (Amos & Kolter, 2017; Donti et al., 2017; Djolonga & Krause, 2017; Tschiatschek et al., 2018; Wilder et al., 2018; Gould et al., 2019). (Interestingly, softmax, sigmoid, and ReLU layers can also be viewed as implicit layers (Amos, 2019), though in practice it is more efficient to use their explicit form.) In principle, such approaches could be used to directly enforce constraints within neural network settings, e.g., by projecting neural network outputs onto a constraint set using quadratic programming layers (Amos & Kolter, 2017) in the case of linear constraints, or convex optimization layers (Agrawal et al., 2019) in the case of general convex constraints. However, given the computational expense of such optimization layers, these projection-based approaches are likely to be inefficient. Instead, our approach leverages insights from this line of work by using implicit differentiation to backpropagate through the "completion" of equality constraints in cases where these constraints cannot be solved explicitly (such as in AC optimal power flow).

## 3 DC3: DEEP CONSTRAINT COMPLETION AND CORRECTION

In this work, we consider solving families of optimization problems for which the objectives and/or constraints vary across instances. Formally, let $x \in \mathbb{R}^d$ denote the problem data, and $y \in \mathbb{R}^n$ denote the solution of the corresponding optimization problem (where $y$ depends on $x$). For any given $x$, our aim is then to find $y$ solving:

$$\underset{y \in \mathbb{R}^n}{\text{minimize}} \ f_x(y), \ \text{s.t.} \ g_x(y) \leq 0, \ h_x(y) = 0, \tag{1}$$

(where $f$, $g$, and $h$ are potentially nonlinear and non-convex). Solving such a family of optimization problems can be framed as a learning problem, where an algorithm must predict an optimal $y$ from the problem data $x$. We consider deep learning approaches to this task – that is, training a neural network $N_\theta$, parameterized by $\theta$, to approximate $y$ given $x$.

A naive deep learning approach to approximating such a problem involves viewing the constraints as a form of regularization. That is, for training examples $x^{(i)}$, the algorithm learns to minimize a composite loss containing both the objective and two "soft loss" terms representing violations of the equality and inequality constraints (for some $\lambda_g, \lambda_h > 0$):

$$\ell_{\text{soft}}(\hat{y}) = f_x(\hat{y}) + \lambda_g \| \text{ReLU}(g_x(\hat{y}))\|_2^2 + \lambda_h \|h_x(\hat{y})\|_2^2. \tag{2}$$

An alternative framework (see, e.g., Zamzam & Baker (2019)) is to use supervised learning on examples $(x^{(i)}, y^{(i)})$ for which an optimum $y^{(i)}$ is known. In this case, the loss is simply $\|\hat{y} - y^{(i)}\|_2^2$. However, both these procedures for training a neural network can lead in practice to highly infeasible outputs (as we demonstrate in our experiments), because they do not strictly enforce constraints. Supervised methods also require constructing a training set (e.g., via an exact solver), a sometimes difficult or expensive step.

To address these challenges, we introduce the method of Deep Constraint Completion and Correction (DC3), which allows hard constraints to be integrated into the training of neural networks. This method is able to train directly from the problem specification (instead of a supervised dataset), via the following two innovations:

**Equality completion.** We provide a mechanism to enforce equality constraints during training and testing, inspired by the literature on variable elimination. Specifically, rather than outputting the full-dimensional optimization solution directly, we first output a subset of the variables, and then infer the remaining variables via the equality constraints – either explicitly, or by solving an implicit set of equations (through which we can then backpropagate via the implicit function theorem).

**Inequality correction.** We correct for violation of the inequality constraints by mapping infeasible points to feasible points using an internal gradient descent procedure during training. This allows us to fix inequality violations while taking steps along the manifold of points satisfying the equalities, which yields an output that is feasible with respect to all constraints.

Overall, our algorithm involves training a neural network $N_\theta(x)$ to output a partial set of variables $z$. These variables are then completed to a full set of variables $\tilde{y}$ satisfying the equality constraints. In turn, $\tilde{y}$ is corrected to $\hat{y}$ to satisfy the inequality constraints while continuing to satisfy the equality constraints. The overall network is trained using backpropagation on the soft loss described in Equation (2) (which is necessary for correction, as noted below). Importantly, both the completion and correction procedures are differentiable either implicitly or explicitly (allowing network training to take them into account), and the overall framework is agnostic to the choice of neural network architecture. A schematic of the DC3 framework is given in Figure 1, and corresponding pseudocode is given in Algorithm 1.

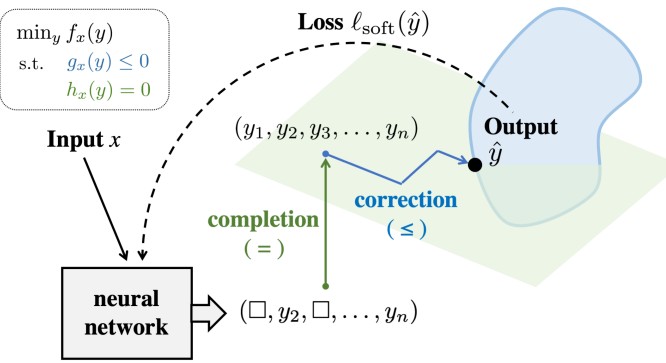

Figure 1: A schematic of the DC3 framework.

---

**Algorithm 1** Deep Constraint Completion and Correction (DC3)

---
1: **assume** equality completion procedure $\varphi_x : \mathbb{R}^m \to \mathbb{R}^{n-m}$ *// to solve equality constraints*
2:
3: **procedure** TRAIN($X$)
4:     **init** neural network $N_\theta : \mathbb{R}^d \to \mathbb{R}^m$
5:     **while** not converged **do** for $x \in X$
6:         **compute** partial set of variables $z = N_\theta(x)$
7:         **complete** to full set of variables $\tilde{y} = \begin{bmatrix} z^T & \varphi_x(z)^T \end{bmatrix}^T \in \mathbb{R}^n$
8:         **correct** to feasible (or approx. feasible) solution $\hat{y} = \rho_x^{(t_{\text{train}})}(\tilde{y})$
9:         **compute** constraint-regularized loss $\ell_{\text{soft}}(\hat{y})$
10:        **update** $\theta$ using $\nabla_\theta \ell_{\text{soft}}(\hat{y})$
11:     **end while**
12: **end procedure**
13:
14: **procedure** TEST($x, N_\theta$)
15:     **compute** partial set of variables $z = N_\theta(x)$
16:     **complete** to full set of variables $\tilde{y} = \begin{bmatrix} z^T & \varphi_x(z)^T \end{bmatrix}^T$
17:     **correct** to feasible solution $\hat{y} = \rho^{(t_{\text{test}})}(\tilde{y})$
18:     **return** $\hat{y}$
19: **end procedure**

---

We note that as this procedure is somewhat general, in cases where constraints have a specialized structure, specialized techniques may be more appropriate to use. For instance, while we examine linearly-constrained settings in our experiments for the purposes of illustration, in practice, techniques such as Minkowski-Weyl decomposition or Cholesky factorization (see Frerix et al. (2020), Amos & Kolter (2017)) may be more efficient in these settings. However, for more general settings without this kind of structure – e.g., non-convex problems such as AC optimal power flow, which we examine in our experiments – the DC3 framework can provide a (differentiable) mechanism for satisfying hard constraints. We now detail the completion and correction procedures used in DC3.

### 3.1 EQUALITY COMPLETION

Assuming that the problem (1) is not overdetermined, the number of equality constraints $h_x(y) = 0$ must not exceed the dimension of the decision variable $y \in \mathbb{R}^n$: that is, the number of equality constraints equals $n - m$ for some $m \geq 0$. Then, given $m$ of the entries of a feasible point $y$, the other $(n-m)$ entries are, in general, determined by the $(n-m)$ equality constraints. We exploit this observation in our algorithm, noting that it is considerably more efficient to output a point in $\mathbb{R}^m$ and complete it to a point $y \in \mathbb{R}^n$ such that $h_x(y) = 0$, as compared with outputting full-dimensional points $y \in \mathbb{R}^n$ and attempting to adjust all coordinates to satisfy the equality constraints.

In particular, we assume that given $m$ entries of $y$, we either can solve for the remaining entries explicitly (e.g. in a linear system) or that we have access to a process (e.g. Newton's Method) allowing us to solve any implicit equations. Formally, we assume access to a function $\varphi_x : \mathbb{R}^m \to \mathbb{R}^{n-m}$ such that $h_x(\begin{bmatrix} z^T & \varphi_x(z)^T \end{bmatrix}^T) = 0$, where $\begin{bmatrix} z^T & \varphi_x(z)^T \end{bmatrix}^T$ is the concatenation of $z$ and $\varphi_x(z)$.

In our algorithm, we then train our neural network $N_\theta$ to output points $z \in \mathbb{R}^m$, which are completed to $\begin{bmatrix} z^T & \varphi_x(z)^T \end{bmatrix}^T \in \mathbb{R}^n$. A challenge then arises as to how to backpropagate the loss during the training of $N_\theta$ if $\varphi_x(z)$ is not a readily differentiable explicit function – for example, if the completion procedure uses Newton's Method. We solve this challenge by leveraging the implicit function theorem, as e.g. in OptNet (Amos & Kolter, 2017) and SATNet (Wang et al., 2019). This approach allows us to express, for any training loss $\ell$, the derivatives $\mathrm{d}\ell/\mathrm{d}z$ using $\mathrm{d}\ell/\mathrm{d}\varphi_x(z)$.

Namely, let $J^h \in \mathbb{R}^{(n-m) \times n}$ denote the Jacobian of $h_x(y)$ with respect to $y$. By the chain rule:

$$0 = \frac{\mathrm{d}}{\mathrm{d}z} h_x\left(\begin{bmatrix} z \\ \varphi_x(z) \end{bmatrix}\right) = \frac{\partial h_x}{\partial z} + \frac{\partial h_x}{\partial \varphi_x(z)}\frac{\partial \varphi_x(z)}{\partial z} = J^h_{:,0:m} + J^h_{:,m:n}\frac{\partial \varphi_x(z)}{\partial z},$$

$$\Rightarrow \quad \partial \varphi_x(z)/\partial z = -\left(J^h_{:,m:n}\right)^{-1} J^h_{:,0:m}. \tag{3}$$

We can then backpropagate losses through the network by noting that

$$\frac{\mathrm{d}\ell}{\mathrm{d}z} = \frac{\partial\ell}{\partial z} + \frac{\partial\ell}{\partial\varphi_x(z)}\frac{\partial\varphi_x(z)}{\partial z} = \frac{\partial\ell}{\partial z} - \frac{\partial\ell}{\partial\varphi_x(z)}\left(J^h_{:,m:n}\right)^{-1}J^h_{:,0:m}. \tag{4}$$

We note that in practice, the Jacobian $\partial\varphi_x(z)/\partial z$ should generally not be computed explicitly due to space complexity considerations; instead, it is often desirable to form the result of the left matrix-vector product $(\partial\ell/\partial\varphi_x(z))(\partial\varphi_x(z)/\partial z)$ directly. This process is well-described in e.g. Amos & Kolter (2017), and also in detail for the problem of AC optimal power flow in Appendix C.

## 3.2 INEQUALITY CORRECTION

While the completion procedure described above guarantees feasibility with respect to the equality constraints, it does not ensure that the inequality constraints will be satisfied. To additionally ensure feasibility with respect to the inequality constraints, our algorithm incorporates a correction procedure that maps the outputs from the previous step into the feasible region. In particular, we employ a gradient-based correction procedure that takes gradient steps in $z$ towards the feasible region *along the manifold of points satisfying the equality constraints*.

Let $\rho_x(y)$ be the operation that takes as input a point $y = \begin{bmatrix} z^T & \varphi_x(z)^T \end{bmatrix}^T$, and moves it closer to satisfying the inequality constraints by taking a step along the gradient of the inequality violation with respect to the partial variables $z$. Formally, for a learning rate $\gamma > 0$, we define:

$$\rho_x\left(\begin{bmatrix} z \\ \varphi_x(z) \end{bmatrix}\right) = \begin{bmatrix} z - \gamma\Delta z \\ \varphi_x(z) - \gamma\Delta\varphi_x(z) \end{bmatrix},$$

$$\text{for} \quad \Delta z = \nabla_z \left\| \mathrm{ReLU}\left( g_x\left(\begin{bmatrix} z \\ \varphi_x(z) \end{bmatrix}\right)\right)\right\|_2^2, \quad \Delta\varphi_x(z) = \frac{\partial\varphi_x(z)}{\partial z}\Delta z.$$

While gradient descent methods do not always converge to global (or local) optima for general optimization problems, if initialized *close to an optimum*, gradient descent is highly effective in practice for non-pathological settings (see e.g. Busseti et al. (2019); Lee et al. (2017)). At test time, the input to the DC3 correction procedure should already be close to feasible with respect to the inequality constraints, as it is the output of a differentiable completion process that is trained using the soft loss $\ell_{\mathrm{soft}}$. Therefore, we may expect that in practice, the limit $\lim_{t\to\infty}\rho_x^{(t)}(y)$ will converge to a point satisfying both inequality and equality constraints (while for problems with linear constraints as in §4.1–4.2, the correction process is mathematically guaranteed to converge).

As the exact limit $\lim_{t\to\infty}\rho_x^{(t)}(y)$ is difficult to calculate in practice, we make approximations at both training and test time. Namely, we apply $\rho_x^{(t)}(y)$ to the output of the completion procedure, with $t = t_{\mathrm{train}}$ relatively small at train time to allow backpropagation through the correction. Depending on time constraints, this same value of $t$ may be used at test time, or a larger value $t = t_{\mathrm{test}} > t_{\mathrm{train}}$ may be used to ensure convergence to a feasible point.

## 4 EXPERIMENTS

We evaluate DC3 for convex quadratic programs (QPs), a simple family of non-convex optimization problems, and the real-world task of AC optimal power flow (ACOPF).[1] In particular, we assess our method on the following criteria:

- **Optimality**: How good is the objective value $f_x(y)$ achieved by the final solution?
- **Feasibility**: How much, if at all, does the solution violate the constraints? Specifically, what are the maximum and mean feasibility violations of the inequality and equality constraints: $\max(\mathrm{ReLU}(g_x(y)))$, $\mathrm{mean}(\mathrm{ReLU}(g_x(y)))$, $\max(h_x(y))$, and $\mathrm{mean}(h_x(y))$?
- **Speed**: How fast is the method?

---

[1] Code for all experiments is available at `https://github.com/locuslab/DC3`

We compare DC3 against the following methods (referred to by abbreviations in our tables below):

- **Optimizer**: A traditional optimization solver. For convex QP settings, we use `OSQP` (Stellato et al., 2020), as well as the batched, differentiable solver `qpth` developed as part of OptNet (Amos & Kolter, 2017). For the generic non-convex setting, we use `IPOPT` (Wächter & Biegler, 2006). For ACOPF, we use the solver provided by `PYPOWER`, a Python port of `MATPOWER` (Zimmerman et al., 1997).

- **NN**: A simple deep learning approach trained to minimize a soft loss (Equation (2)).

- **NN, ≤ test**: The NN approach, with a gradient-based correction procedure[2] applied to the output at test time in an effort to mitigate violations of equality and inequality constraints. Unlike in DC3, correction is not used during training, and completion is not used at all.

- **Eq. NN**: A more sophisticated approach inspired by[3] that in Zamzam & Baker (2019), where (i) the neural network outputs a partial set of variables $\hat{z}$, which is completed to the full set using the equality constraints, (ii) training is performed by supervised learning on optimal pairs $(x^{(i)}, z^{(i)})$ with loss function $||\hat{z} - z^{(i)}||_2^2$, not using the objective value at all.

- **Eq. NN, ≤ test**: The approach in Eq. NN, augmented with gradient-based correction at test time to mitigate violations of equality and inequality constraints.

We also attempted to use the output of the **NN** method as a "warm start" for traditional optimizers, but found that the **NN** output was sufficiently far from feasibility that it did not help.

In addition, we consider weaker versions of DC3 in which components of the algorithm are ablated:

- **DC3, ≠**. The DC3 algorithm with completion ablated. All variables are output by the network directly and correction is performed by taking gradient steps for both equality and inequality constraints.

- **DC3, ⪇ train**. The DC3 algorithm with correction ablated at train time. Correction is still performed at test time.

- **DC3, ⪇ train/test**. The DC3 algorithm with correction ablated at both train and test time.

- **DC3, no soft loss**. The DC3 algorithm with training performed to minimize the objective value only, without auxiliary terms capturing equality and inequality violation.

As our overall framework is agnostic to the choice of neural network architecture, to facilitate comparison, we use a fixed neural network architecture across all experiments: fully connected with two hidden layers of size 200, including ReLU activation, batch normalization, and dropout (with rate 0.2) at each hidden layer (Ioffe & Szegedy, 2015; Srivastava et al., 2014). For our correction procedure, we use $t_{\text{train}} = t_{\text{test}} = 10$ for the convex QP and simple non-convex tasks and $t_{\text{train}} = t_{\text{test}} = 5$ for ACOPF (see Appendix B). All neural networks are trained using PyTorch (Paszke et al., 2019).

To generate timing results, all neural nets and the `qpth` optimizer were run with full parallelization on a GeForce GTX 2080 Ti GPU. The `OSQP`, `IPOPT`, and `PYPOWER` optimizers were run sequentially on an Intel Xeon 2.10GHz CPU, and we report the total time divided by the number of test instances to simulate full parallelization. As our implementations are not tightly optimized, we emphasize that all timing comparisons are approximate.

## 4.1 CONVEX QUADRATIC PROGRAMS

As a first test of the DC3 method, we consider solving convex quadratic programs with a quadratic objective function and linear constraints. Note that we examine this simple task first for illustration,

---

[2]Note that this correction procedure is not exactly the same as that described in Section 3.2, as the outputs of the NN baseline do not necessarily meet the prerequisite of satisfying the equality constraints. Instead, we adjust the full set of output variables directly with respect to gradients of the inequality and equality violations.

[3]In Zamzam & Baker (2019), the authors employ one step of an ACOPF-specific heuristic called PV/PQ switching to correct inequality constraint violations at test time. We do not apply this heuristic here in the spirit of presenting a more general framework. As PV/PQ switching is not necessarily guaranteed to correct all inequality violations (although it can work well in practice), in principle, one could consider employing a combination of PV/PQ switching *and* gradient-based corrections in the context of ACOPF.

| | Obj. value | Max eq. | Mean eq. | Max ineq. | Mean ineq. | Time (s) |
|---|---|---|---|---|---|---|
| Optimizer (OSQP) | -15.05 (0.00) | 0.00 (0.00) | 0.00 (0.00) | 0.00 (0.00) | 0.00 (0.00) | 0.002 (0.000) |
| Optimizer (qpth) | -15.05 (0.00) | 0.00 (0.00) | 0.00 (0.00) | 0.00 (0.00) | 0.00 (0.00) | 1.335 (0.012) |
| DC3 | -13.46 (0.01) | 0.00 (0.00) | 0.00 (0.00) | 0.00 (0.00) | 0.00 (0.00) | 0.017 (0.001) |
| DC3, $\neq$ | -12.58 (0.04) | 0.35 (0.00) | 0.13 (0.00) | 0.00 (0.00) | 0.00 (0.00) | 0.008 (0.000) |
| DC3, $\not\leq$ train | -1.39 (0.97) | 0.00 (0.00) | 0.00 (0.00) | 0.02 (0.02) | 0.00 (0.00) | 0.017 (0.000) |
| DC3, $\not\leq$ train/test | -1.23 (1.21) | 0.00 (0.00) | 0.00 (0.00) | 0.09 (0.13) | 0.01 (0.01) | 0.001 (0.000) |
| DC3, no soft loss | -21.84 (0.00) | 0.00 (0.00) | 0.00 (0.00) | 23.83 (0.11) | 4.04 (0.01) | 0.017 (0.000) |
| NN | -12.57 (0.01) | 0.35 (0.00) | 0.13 (0.00) | 0.00 (0.00) | 0.00 (0.00) | 0.001 (0.000) |
| NN, $\leq$ test | -12.57 (0.01) | 0.35 (0.00) | 0.13 (0.00) | 0.00 (0.00) | 0.00 (0.00) | 0.008 (0.000) |
| Eq. NN | -9.16 (0.75) | 0.00 (0.00) | 0.00 (0.00) | 8.83 (0.72) | 0.91 (0.09) | 0.001 (0.000) |
| Eq. NN, $\leq$ test | -14.68 (0.05) | 0.00 (0.00) | 0.00 (0.00) | 0.89 (0.05) | 0.07 (0.01) | 0.018 (0.001) |

Table 1: Results on QP task for 100 variables, 50 equality constraints, and 50 inequality constraints. We compare the performance of DC3 and other algorithms according to the objective value and max/mean values of equality/inequality constraint violations, each averaged across test instances. We also compare the total time required to run on all 833 test instances, assuming full parallelization. (Std. deviations across 5 runs are shown in parentheses for all figures reported.) Lower values are better for all metrics. We find that methods other than DC3 and Optimizer violate feasibility (as shown in red). DC3 gives a feasible output with reasonable objective value $78\times$ faster than qpth and only $9\times$ slower than OSQP, which is optimized for convex QPs.

but the general DC3 method is assuredly overkill for solving convex quadratic programs. It may not even be the most efficient deep learning-based method for constraint enforcement on this task, since more specialized techniques are available in such linearly constrained settings (Frerix et al., 2020).

We consider the following problem:

$$\underset{y \in \mathbb{R}^n}{\text{minimize}} \ \frac{1}{2} y^T Q y + p^T y, \ \text{ s. t. } Ay = x, \ Gy \leq h, \tag{5}$$

for constants $Q \in \mathbb{R}^{n \times n} \succeq 0$, $p \in \mathbb{R}^n$, $A \in \mathbb{R}^{n_{\text{eq}} \times n}$, $G \in \mathbb{R}^{n_{\text{ineq}} \times n}$, $h \in \mathbb{R}^{n_{\text{ineq}}}$, and variable $x \in \mathbb{R}^{n_{\text{eq}}}$ which varies between problem instances. We must learn to approximate the optimal $y$ given $x$.

In our experiments, we take $Q$ to be a diagonal matrix with all diagonal entries drawn i.i.d. from the uniform distribution on $[0, 1]$, ensuring that $Q$ is positive semi-definite. We take matrices $A, G$ with entries drawn i.i.d. from the unit normal distribution. We assume that in each problem instance, all entries of $x$ are in the interval $[-1, 1]$. In order to ensure that the problem is feasible, we take $h = \sum_j |(GA^+)_{ij}|$, where $A^+$ is the Moore-Penrose pseudoinverse of $A$; namely, for this choice of $h$, the point $y = A^+ x$ is feasible (but not, in general, optimal), because:

$$AA^+ x = x, \qquad GA^+ x \leq \sum_j \left| (GA^+)_{ij} \right| \text{ since } |x_j| \leq 1. \tag{6}$$

During training, we use examples $x$ with entries drawn i.i.d. from the uniform distribution on $[-1, 1]$.

Table 1 compares the performance of DC3 (and various ablations of DC3) with traditional optimizers and other deep learning-based methods, for the case of $n = 100$ variables and $n_{\text{eq}} = n_{\text{ineq}} = 50$. In Appendix A, we evaluate settings with other numbers of equality and inequality constraints. Each experiment is run 5 times for 10,000 examples $x$ (with train/test/validation ratio 10:1:1). Hyperparameters are tuned to maximize performance for each method individually (see Appendix B).

We find that DC3 preserves feasibility with respect to both equality and inequality constraints, while achieving reasonable objective values. (The average per-instance optimality gap for DC3 over the classical optimizer is 10.59%.) For every baseline deep learning algorithm, on the other hand, feasibility is violated significantly for either equality or inequality constraints. As expected, "DC3 $\neq$" (completion ablated) results in violated equality constraints, while "DC3 $\not\leq$" (correction ablated) violates inequality constraints. Ablating the soft loss also results in violated inequality constraints, leading to an objective value significantly lower than would be possible were constraints satisfied.

Even though we have not optimized the code of DC3 to be maximally fast, our implementation of DC3 still runs about $78\times$ faster than the state-of-the-art differentiable QP solver qpth, and only

|  | Obj. value | Max eq. | Mean eq. | Max ineq. | Mean ineq. | Time (s) |
|---|---|---|---|---|---|---|
| Optimizer | -11.59 (0.00) | 0.00 (0.00) | 0.00 (0.00) | 0.00 (0.00) | 0.00 (0.00) | 0.121 (0.000) |
| DC3 | -10.66 (0.03) | 0.00 (0.00) | 0.00 (0.00) | 0.00 (0.00) | 0.00 (0.00) | 0.013 (0.000) |
| DC3, $\neq$ | -10.04 (0.02) | 0.35 (0.00) | 0.13 (0.00) | 0.00 (0.00) | 0.00 (0.00) | 0.009 (0.000) |
| DC3, $\not\leq$ train | -0.29 (0.67) | 0.00 (0.00) | 0.00 (0.00) | 0.01 (0.01) | 0.00 (0.00) | 0.010 (0.004) |
| DC3, $\not\leq$ train/test | -0.27 (0.67) | 0.00 (0.00) | 0.00 (0.00) | 0.03 (0.03) | 0.00 (0.00) | 0.001 (0.000) |
| DC3, no soft loss | -13.81 (0.00) | 0.00 (0.00) | 0.00 (0.00) | 15.21 (0.04) | 2.33 (0.01) | 0.013 (0.000) |
| NN | -10.02 (0.01) | 0.35 (0.00) | 0.13 (0.00) | 0.00 (0.00) | 0.00 (0.00) | 0.001 (0.000) |
| NN, $\leq$ test | -10.02 (0.01) | 0.35 (0.00) | 0.13 (0.00) | 0.00 (0.00) | 0.00 (0.00) | 0.009 (0.000) |
| Eq. NN | -3.88 (0.56) | 0.00 (0.00) | 0.00 (0.00) | 6.87 (0.43) | 0.72 (0.05) | 0.001 (0.000) |
| Eq. NN, $\leq$ test | -10.99 (0.03) | 0.00 (0.00) | 0.00 (0.00) | 0.87 (0.04) | 0.06 (0.00) | 0.013 (0.000) |

Table 2: Results on our simple nonconvex task for 100 variables, 50 equality constraints, and 50 inequality constraints, with details as in Table 1. Since this problem is nonconvex, we use IPOPT as the classical optimizer. DC3 is differentiable and about 9× faster than IPOPT, giving a near-optimal objective value and constraint satisfaction, in contrast to baseline deep learning-based methods which result in significant constraint violations.

about 9× slower than the classical optimizer OSQP, which is specifically optimized for convex QPs. Furthermore, this assumes OSQP is fully parallelized – in this case, across 833 CPUs – whereas standard, non-parallel implementations of OSQP would be orders of magnitude slower. By contrast, DC3 is easily parallelized within a single GPU using standard deep learning frameworks.

## 4.2 SIMPLE NON-CONVEX OPTIMIZATION

We now consider a simple non-convex adaptation of the quadratic program above:

$$\underset{y \in \mathbb{R}^n}{\text{minimize}} \ \frac{1}{2}y^T Q y + p^T \sin(y), \ \text{s.t.} \ Ay = x, \ Gy \leq h,$$

where $\sin(y)$ denotes the componentwise application of the sine function to the vector $y$, and where all constants and variables are defined as in (5). We consider instances of this problem where all parameters are drawn randomly as in our preceding experiments in the convex setting.

In Table 2, we compare the performance of DC3 and other deep learning-based methods against the classical non-convex optimizer IPOPT.[4] We find that DC3 achieves good objective values (8.02% per-instance optimality gap), while maintaining feasibility. By contrast, all other deep learning-based methods that we consider violate constraints significantly. DC3 also runs about 10× faster than IPOPT, even assuming IPOPT is fully parallelized. (Even on the CPU, DC3 takes $0.030 \pm 0.000$ seconds, about 4× faster than IPOPT.) Note that the DC3 algorithm is essentially the same between the convex QP and this non-convex task, since only the objective function is altered.

## 4.3 AC OPTIMAL POWER FLOW

We now show how DC3 can be applied to the problem of AC optimal power flow (ACOPF). ACOPF is a fundamental problem for the operation of the electrical grid, and is used to determine how much power must be produced by each generator on the grid in order to meet demand. As the amount of renewable energy on the power grid grows, this problem must be solved more and more frequently to account for the variability of these renewable sources, and at larger scale to account for an increasing number of distributed devices (Rolnick et al., 2019). However, ACOPF is a non-convex optimization problem and classical optimizers scale poorly on it. While specialized approaches to this problem have started to emerge, including using machine learning (see, e.g., Zamzam & Baker (2019) for a discussion), we here assess the ability of our more general framework to address this problem.

Formally, a power network may be considered as a graph on $b$ nodes, representing different locations (*buses*) within the electrical grid, and with edges weighted by complex numbers $w_{ij} \in \mathbb{C}$ (*admittances*) that represent how easily current can flow between the corresponding locations in the grid. Let $W \in \mathbb{C}^{b \times b}$ denote the graph Laplacian (or *nodal admittance matrix*). Then, the problem of ACOPF can be defined as follows: Given input variables $p_d \in \mathbb{R}^b$, $q_d \in \mathbb{R}^b$ (representing *real power*

---

[4]We initialize the optimizer using the feasible point $y = A^+ x$ noted in Equation (6).

|  | Obj. value | Max eq. | Mean eq. | Max ineq. | Mean ineq. | Time (s) |
|---|---|---|---|---|---|---|
| Optimizer | 3.81 (0.00) | 0.00 (0.00) | 0.00 (0.00) | 0.00 (0.00) | 0.00 (0.00) | 0.949 (0.002) |
| DC3 | 3.82 (0.00) | 0.00 (0.00) | 0.00 (0.00) | 0.00 (0.00) | 0.00 (0.00) | 0.089 (0.000) |
| DC3, $\neq$ | 3.67 (0.01) | 0.14 (0.01) | 0.02 (0.00) | 0.00 (0.00) | 0.00 (0.00) | 0.040 (0.000) |
| DC3, $\not\leq$ train | 3.82 (0.00) | 0.00 (0.00) | 0.00 (0.00) | 0.00 (0.00) | 0.00 (0.00) | 0.089 (0.000) |
| DC3, $\not\leq$ train/test | 3.82 (0.00) | 0.00 (0.00) | 0.00 (0.00) | 0.01 (0.00) | 0.00 (0.00) | 0.039 (0.000) |
| DC3, no soft loss | 3.11 (0.05) | 2.60 (0.35) | 0.07 (0.00) | 2.33 (0.33) | 0.03 (0.01) | 0.088 (0.000) |
| NN | 3.69 (0.02) | 0.19 (0.01) | 0.03 (0.00) | 0.00 (0.00) | 0.00 (0.00) | 0.001 (0.000) |
| NN, $\leq$ test | 3.69 (0.02) | 0.16 (0.00) | 0.02 (0.00) | 0.00 (0.00) | 0.00 (0.00) | 0.040 (0.000) |
| Eq. NN | 3.81 (0.00) | 0.00 (0.00) | 0.00 (0.00) | 0.15 (0.01) | 0.00 (0.00) | 0.039 (0.000) |
| Eq. NN, $\leq$ test | 3.81 (0.00) | 0.00 (0.00) | 0.00 (0.00) | 0.15 (0.01) | 0.00 (0.00) | 0.078 (0.000) |

Table 3: Results on ACOPF over 100 test instances. We compare the performance of DC3 and other algorithms according to the metrics described in Table 1. We find again that baseline methods violate feasibility (as shown in red), while DC3 gives a feasible and near-optimal output about $10\times$ faster than the PYPOWER optimizer, even assuming that PYPOWER is fully parallelized.

and *reactive power demand* at the various nodes of the graph), output the variables $p_g \in \mathbb{R}^b, q_g \in \mathbb{R}^b$ (representing real power and reactive power *generation*) and $v \in \mathbb{C}^b$ (representing *real* and *imaginary voltage*), according to the following optimization problem:

$$\underset{p_g \in \mathbb{R}^b,\, q_g \in \mathbb{R}^b,\, v \in \mathbb{C}^b}{\text{minimize}} \quad p_g^T A p_g + b^T p_g$$
$$\text{subject to} \quad p_g^{\min} \leq p_g \leq p_g^{\max}, \quad q_g^{\min} \leq q_g \leq q_g^{\max}, \quad v^{\min} \leq |v| \leq v^{\max}, \tag{7}$$
$$(p_g - p_d) + (q_g - q_d)i = \text{diag}(v)\overline{W}\overline{v}.$$

More details about how we apply DC3 to the problem of ACOPF are given in Appendix C.

We assess our method on a 57-node power system test case available via the MATPOWER package. We conduct 5 runs over 1,200 input datapoints (with a train/test/validation ratio of 10:1:1). As with other tasks, hyperparameters for ACOPF were tuned to maximize performance for each method individually (see Appendix B). Optimality, feasibility, and timing results are reported in Table 3.

We find that DC3 achieves comparable objective values to the optimizer, and preserves feasibility with respect to both equality and inequality constraints. Once again, for every baseline deep learning algorithm, feasibility is violated significantly for either equality or inequality constraints. Ablations of DC3 also suffer from constraint violations, though the effect is less pronounced than for the convex QP and simple non-convex settings, especially for ablation of the correction (perhaps because the inequality constraints here are easier to satisfy than the equality constraints). We also see that DC3 runs about $10\times$ faster than the PYPOWER optimizer, even when PYPOWER is fully parallelized. (Even when running on the CPU, DC3 takes $0.906 \pm 0.003$ seconds, slightly faster than PYPOWER.)

Out of 100 test instances, there were 3 on which DC3 output lower-than-optimal objective values of up to a few percent (-0.30%, -1.85%, -5.51%), reflecting slight constraint violations. Over the other 97 instances, the per-instance optimality gap compared to the classical optimizer was 0.22%.

## 5   CONCLUSION

We have described a method, DC3, for fast approximate solutions to optimization problems with hard constraints. Our approach includes a neural network that outputs a partial set of variables, a differentiable completion procedure that fills in remaining variables according to equality constraints, and a differentiable correction procedure that fixes inequality violations. We find that DC3 yields solutions of significantly better feasibility and objective value than other approximate deep learning-based solvers on convex and non-convex optimization tasks.

We note that, while DC3 provides a general framework for tackling constrained optimization, depending on the setting, the expensiveness of both the completion and correction procedures may vary (e.g., implicit solutions may be more time-consuming, or gradient descent may converge more or less easily). We believe that, while our method as stated is broadly applicable, it will be possible in future work to design further improvements tailored to specific problem instances, for example by designing problem-dependent correction procedures.

ACKNOWLEDGMENTS

This work was supported by the U.S. Department of Energy Computational Science Graduate Fellowship (DE-FG02-97ER25308), U.S. National Science Foundation (DMS-1803547), the Center for Climate and Energy Decision Making through a cooperative agreement between the National Science Foundation and Carnegie Mellon University (SES-00949710), the Computational Sustainability Network, and the Bosch Center for AI.

We thank Shaojie Bai, Rizal Fathony, Filipe de Avila Belbute Peres, Josh Williams, and anonymous reviewers for their feedback on this work.

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

| | | | | | | |
|---|---|---|---|---|---|---|
| Optimizers (OSQP, qpth) | Obj. val. | -27.26 (0.00) | -23.13 (0.00) | -15.05 (0.00) | -14.80 (0.00) | -4.79 (0.00) |
| | Max eq. | 0.00 (0.00) | 0.00 (0.00) | 0.00 (0.00) | 0.00 (0.00) | 0.00 (0.00) |
| | Max ineq. | 0.00 (0.00) | 0.00 (0.00) | 0.00 (0.00) | 0.00 (0.00) | 0.00 (0.00) |
| DC3 | Obj. val. | -25.79 (0.02) | -20.29 (0.20) | -13.46 (0.01) | -13.73 (0.02) | -4.76 (0.00) |
| | Max eq. | 0.00 (0.00) | 0.00 (0.00) | 0.00 (0.00) | 0.00 (0.00) | 0.00 (0.00) |
| | Max ineq. | 0.00 (0.00) | 0.00 (0.00) | 0.00 (0.00) | 0.00 (0.00) | 0.00 (0.00) |
| DC3, $\neq$ | Obj. val. | -22.59 (0.18) | -20.40 (0.03) | -12.58 (0.04) | -13.36 (0.02) | -5.27 (0.00) |
| | Max eq. | 0.12 (0.01) | 0.24 (0.00) | 0.35 (0.00) | 0.47 (0.00) | 0.59 (0.00) |
| | Max ineq. | 0.00 (0.00) | 0.00 (0.00) | 0.00 (0.00) | 0.00 (0.00) | 0.00 (0.00) |
| DC3, $\not\leq$ train | Obj. val. | -25.75 (0.04) | -20.14 (0.15) | -1.39 (0.97) | -13.71 (0.04) | -4.75 (0.00) |
| | Max eq. | 0.00 (0.00) | 0.00 (0.00) | 0.00 (0.00) | 0.00 (0.00) | 0.00 (0.00) |
| | Max ineq. | 0.00 (0.00) | 0.00 (0.00) | 0.02 (0.02) | 0.00 (0.00) | 0.00 (0.00) |
| DC3, $\not\leq$ train/test | Obj. val. | -25.75 (0.04) | -20.14 (0.15) | -1.23 (1.21) | -13.71 (0.04) | -4.75 (0.00) |
| | Max eq. | 0.00 (0.00) | 0.00 (0.00) | 0.00 (0.00) | 0.00 (0.00) | 0.00 (0.00) |
| | Max ineq. | 0.00 (0.00) | 0.00 (0.00) | 0.09 (0.13) | 0.00 (0.00) | 0.00 (0.00) |
| DC3, no soft loss | Obj. val. | -66.79 (0.01) | -40.63 (0.02) | -21.84 (0.00) | -15.56 (0.02) | -4.76 (0.01) |
| | Max eq. | 0.00 (0.00) | 0.00 (0.00) | 0.00 (0.00) | 0.00 (0.00) | 0.00 (0.00) |
| | Max ineq. | 122.52 (0.22) | 50.25 (0.14) | 23.83 (0.11) | 5.85 (0.18) | 0.00 (0.00) |
| NN | Obj. val. | -22.65 (0.12) | -20.43 (0.03) | -12.57 (0.01) | -13.35 (0.03) | -5.29 (0.02) |
| | Max eq. | 0.11 (0.01) | 0.24 (0.00) | 0.35 (0.00) | 0.47 (0.00) | 0.59 (0.00) |
| | Max ineq. | 0.00 (0.00) | 0.00 (0.00) | 0.00 (0.00) | 0.00 (0.00) | 0.00 (0.00) |
| NN, $\leq$ test | Obj. val. | -22.65 (0.12) | -20.43 (0.03) | -12.57 (0.01) | -13.35 (0.03) | -5.29 (0.02) |
| | Max eq. | 0.11 (0.01) | 0.24 (0.00) | 0.35 (0.00) | 0.47 (0.00) | 0.59 (0.00) |
| | Max ineq. | 0.00 (0.00) | 0.00 (0.00) | 0.00 (0.00) | 0.00 (0.00) | 0.00 (0.00) |
| Eq. NN | Obj. val. | -27.20 (0.02) | -22.74 (0.14) | -9.16 (0.75) | -14.64 (0.01) | -4.74 (0.01) |
| | Max eq. | 0.00 (0.00) | 0.00 (0.00) | 0.00 (0.00) | 0.00 (0.00) | 0.00 (0.00) |
| | Max ineq. | 0.43 (0.02) | 1.37 (0.09) | 8.83 (0.72) | 0.86 (0.05) | 0.00 (0.00) |
| Eq. NN, $\leq$ test | Obj. val. | -27.20 (0.02) | -22.76 (0.13) | -14.68 (0.05) | -14.65 (0.01) | -4.74 (0.01) |
| | Max eq. | 0.00 (0.00) | 0.00 (0.00) | 0.00 (0.00) | 0.00 (0.00) | 0.00 (0.00) |
| | Max ineq. | 0.42 (0.02) | 1.27 (0.09) | 0.89 (0.05) | 0.85 (0.05) | 0.00 (0.00) |

Table A.1: Results on QP task for 100 variables and 50 inequality constraints, as the number of equality constraints varies as $10, 30, 50, 70, 90$. We compare the performance of DC3 and other algorithms according to objective value and maximum equality/inequality constraint violations averaged across test instances. (Standard deviations across 5 runs are shown in parentheses.) We find that neural network baselines give significantly inferior performance (shown in red) across all problems.

# A   ADDITIONAL QP TASK RESULTS

We compare the performance of DC3 against other methods for the convex QP task with 100 variables, as the number of equality constraints (Table A.1) and inequality constraints (Table A.2) varies.

# B   DETAILS ON HYPERPARAMETER TUNING

The following parameters were kept fixed for all neural network-based methods across all experiments, based on a small amount of informal experimentation to ensure training was stable and properly converged:

- Epochs: 1000
- Batch size:[5] 200
- Hidden layer size: 200 (for both hidden layers)
- Correction procedure stopping tolerance: $10^{-4}$
- Correction procedure momentum: 0.5

---

[5]While this batch size was used for training, final timing experiments were run with all test datapoints in one batch.

|  |  | 10 | 30 | 50 | 70 | 90 |
|---|---|---|---|---|---|---|
| Optimizers (OSQP, qpth) | Obj. val. | -17.33 (0.00) | -16.33 (0.00) | -15.05 (0.00) | -14.61 (0.00) | -14.26 (0.00) |
|  | Max eq. | 0.00 (0.00) | 0.00 (0.00) | 0.00 (0.00) | 0.00 (0.00) | 0.00 (0.00) |
|  | Max ineq. | 0.00 (0.00) | 0.00 (0.00) | 0.00 (0.00) | 0.00 (0.00) | 0.00 (0.00) |
| DC3 | Obj. val. | -15.18 (0.31) | -13.90 (0.20) | -13.46 (0.01) | -10.52 (0.93) | -11.77 (0.07) |
|  | Max eq. | 0.00 (0.00) | 0.00 (0.00) | 0.00 (0.00) | 0.00 (0.00) | 0.00 (0.00) |
|  | Max ineq. | 0.00 (0.00) | 0.00 (0.00) | 0.00 (0.00) | 0.00 (0.00) | 0.00 (0.00) |
| DC3, $\neq$ | Obj. val. | -15.66 (0.05) | -14.10 (0.04) | -12.58 (0.04) | -12.10 (0.03) | -11.72 (0.03) |
|  | Max eq. | 0.35 (0.00) | 0.35 (0.00) | 0.35 (0.00) | 0.35 (0.00) | 0.35 (0.00) |
|  | Max ineq. | 0.00 (0.00) | 0.00 (0.00) | 0.00 (0.00) | 0.00 (0.00) | 0.00 (0.00) |
| DC3, $\not\leq$ train | Obj. val. | -15.60 (0.30) | -13.83 (0.15) | -1.39 (0.97) | -11.14 (0.05) | -11.76 (0.06) |
|  | Max eq. | 0.00 (0.00) | 0.00 (0.00) | 0.00 (0.00) | 0.00 (0.00) | 0.00 (0.00) |
|  | Max ineq. | 0.00 (0.00) | 0.00 (0.00) | 0.02 (0.02) | 0.00 (0.00) | 0.00 (0.00) |
| DC3, $\not\leq$ train/test | Obj. val. | -15.60 (0.30) | -13.83 (0.15) | -1.23 (1.21) | -11.14 (0.05) | -11.76 (0.06) |
|  | Max eq. | 0.00 (0.00) | 0.00 (0.00) | 0.00 (0.00) | 0.00 (0.00) | 0.00 (0.00) |
|  | Max ineq. | 0.00 (0.00) | 0.00 (0.00) | 0.09 (0.13) | 0.00 (0.00) | 0.00 (0.00) |
| DC3, no soft loss | Obj. val. | -21.78 (0.01) | -21.72 (0.03) | -21.84 (0.00) | -21.82 (0.01) | -21.83 (0.00) |
|  | Max eq. | 0.00 (0.00) | 0.00 (0.00) | 0.00 (0.00) | 0.00 (0.00) | 0.00 (0.00) |
|  | Max ineq. | 23.85 (0.52) | 23.54 (0.23) | 23.83 (0.11) | 23.73 (0.09) | 23.69 (0.06) |
| NN | Obj. val. | -15.66 (0.03) | -14.10 (0.05) | -12.57 (0.01) | -12.12 (0.03) | -11.70 (0.02) |
|  | Max eq. | 0.35 (0.00) | 0.35 (0.00) | 0.35 (0.00) | 0.35 (0.00) | 0.35 (0.00) |
|  | Max ineq. | 0.00 (0.00) | 0.00 (0.00) | 0.00 (0.00) | 0.00 (0.00) | 0.00 (0.00) |
| NN, $\leq$ test | Obj. val. | -15.66 (0.03) | -14.10 (0.05) | -12.57 (0.01) | -12.12 (0.03) | -11.70 (0.02) |
|  | Max eq. | 0.35 (0.00) | 0.35 (0.00) | 0.35 (0.00) | 0.35 (0.00) | 0.35 (0.00) |
|  | Max ineq. | 0.00 (0.00) | 0.00 (0.00) | 0.00 (0.00) | 0.00 (0.00) | 0.00 (0.00) |
| Eq. NN | Obj. val. | -17.07 (0.20) | -15.45 (0.09) | -9.16 (0.75) | -14.20 (0.06) | -14.10 (0.10) |
|  | Max eq. | 0.00 (0.00) | 0.00 (0.00) | 0.00 (0.00) | 0.00 (0.00) | 0.00 (0.00) |
|  | Max ineq. | 1.65 (0.22) | 1.79 (0.10) | 8.83 (0.72) | 2.26 (0.04) | 1.28 (0.05) |
| Eq. NN, $\leq$ test | Obj. val. | -17.06 (0.20) | -15.65 (0.09) | -14.68 (0.05) | -14.31 (0.06) | -14.10 (0.10) |
|  | Max eq. | 0.00 (0.00) | 0.00 (0.00) | 0.00 (0.00) | 0.00 (0.00) | 0.00 (0.00) |
|  | Max ineq. | 1.53 (0.19) | 1.58 (0.08) | 0.89 (0.05) | 1.92 (0.04) | 1.25 (0.04) |

Table A.2: Results on QP task for 100 variables and 50 equality constraints. The number of *inequality* constraints varies as $10, 30, 50, 70, 90$. Content and interpretation as in Table A.1.

For all remaining parameters, we performed hyperparameter tuning via a coordinate search over relevant parameters. Central values for this coordinate search were determined via a small amount of informal experimentation to ensure training was stable on that central set of parameters. Final parameters were chosen so as to prioritize feasibility (as determined via the mean and max violations of equality and inequality constraints), followed by objective value and speed.

### B.1 CONVEX QUADRATIC PROGRAMS

We list the ranges over which different hyperparameters were tuned for each method, with central values for the coordinate search in *italics*, and the final parameter values in **bold**. In order to minimize the amount of tuning, we employ the hyperparameters obtained for DC3 to "DC3, $\not\leq$ train" and "DC3, no soft loss" as well (rather than tuning the latter two methods separately). For this set of experiments, we set the learning rate of the gradient-based correction procedure $\rho_x$ to $10^{-7}$ for all methods, as most methods went unstable for higher correction learning rates.

|  | DC3
DC3, $\not\leq$ train
DC3, no soft loss | DC3, $\neq$ | NN | Eq. NN |
|---|---|---|---|---|
| Lr | $10^{-2}, 10^{-3}, \mathbf{10^{-4}}$ | $10^{-2}, 10^{-3}, \mathbf{10^{-4}}$ | $10^{-2}, 10^{-3}, \mathbf{10^{-4}}$ | $10^{-2}, 10^{-3}, \mathbf{10^{-4}}$ |
| $\lambda_g + \lambda_h$ | 1,**10**,100 | 1,*10*,**100** | 1,*10*,**100** | – |
| $\frac{\lambda_h}{\lambda_g + \lambda_h}$ | **0.5**, 0.1, 0.01 | **0.5**, 0.9, 0.99 | **0.5**, 0.9, 0.99 | – |
| $t_{\text{test}}$ | 5, **10**, 50, 100, *1000* | 5, **10**, 50, 100, *1000* | 5, **10**, 50, 100, *1000* | 5, **10**, 50, 100, *1000* |
| $t_{\text{train}}$ | 1, 2, *5*, **10**, 100 | 1, 2, *5*, **10**, 100 | – | – |

## B.2 SIMPLE NON-CONVEX OPTIMIZATION

All hyperparameters in these experiments were maintained at the settings chosen for the convex QP task, since these tasks were similar by design.

## B.3 AC OPTIMAL POWER FLOW

We list the ranges over which different hyperparameters were tuned for each method, with central values for the coordinate search in *italics*, and the final parameter values in **bold**. As in the previous setting, in order to minimize the amount of tuning, we employ the hyperparameters obtained for DC3 to "DC3, $\not\leq$ train" and "DC3, no soft loss" as well (rather than tuning the latter two methods separately).

| | DC3
DC3, $\not\leq$ train
DC3, no soft loss | DC3, $\neq$ | NN | Eq. NN |
|---|---|---|---|---|
| Lr | $10^{-2}$, $\mathbf{\mathit{10^{-3}}}$, $10^{-4}$ | $10^{-2}$, $\mathbf{\mathit{10^{-3}}}$, $10^{-4}$ | $10^{-2}$, $\mathbf{\mathit{10^{-3}}}$, $10^{-4}$ | $10^{-2}$, $\mathbf{\mathit{10^{-3}}}$, $10^{-4}$ |
| $\lambda_g + \lambda_h$ | 1,*10*,100 | 1,*10*,**100** | 1,*10*,**100** | – |
| $\frac{\lambda_h}{\lambda_g+\lambda_h}$ | **0.5**, 0.1, 0.01 | **0.5**, 0.9, 0.99 | **0.5**, 0.9, 0.99 | – |
| $t_{\text{test}}$ | **5**, *10*, 50, 100 | **5**, *10*, 50, 100 | **5**, *10*, 50, 100 | **5**, *10*, 50, 100, 200 |
| $t_{\text{train}}$ | 1, 2, **5**, 10, 100 | 1, 2, **5**, 10, 100 | – | – |
| Lr, $\rho_x$ | $10^{-3}$, $\mathbf{\mathit{10^{-4}}}$, $10^{-5}$ | $10^{-4}$, $\mathbf{\mathit{10^{-5}}}$, $10^{-6}$ | $\mathbf{10^{-5}}$, $\mathit{10^{-6}}$, $10^{-7}$ | $\mathbf{10^{-5}}$, $\mathit{10^{-6}}$, $10^{-7}$ |

# C DETAILS OF DC3 FOR ACOPF

## C.1 PROBLEM SETTING

We consider the problem of ACOPF defined in §4.3.

Let $\mathcal{B}$ denote the overall set of buses (i.e., nodes in the power network). In any instance of ACOPF there exists a set $\mathcal{D} \subseteq \mathcal{B}$ of *load (demand) buses* at which $p_g$ and $q_g$ are identically zero, as well as a set $\mathcal{R} \subseteq \mathcal{B}$ of *reference (slack) buses* at which $p_g$ and $q_g$ are potentially nonzero, and where the voltage angle $\angle v$ is known. Let $\mathcal{G} = \mathcal{B} \setminus (\mathcal{D} \cup \mathcal{R})$ be the remaining *generator buses* at which $p_g$ and $q_g$ are potentially nonzero, but where the voltage angle $\angle v$ is not known.

Then, we may rewrite Equation (7) as follows, where $v \equiv |v|e^{i\angle v}$ and $W \equiv W_r + W_i i$:

$$\underset{p_g,q_g,|v|,\angle v\in\mathbb{R}^{|\mathcal{B}|}}{\text{minimize}} \quad p_g^T A p_g + b^T p_g \tag{C.1a}$$

$$\text{subject to} \quad p_g^{\min} \leq p_g \leq p_g^{\max} \tag{C.1b}$$

$$q_g^{\min} \leq q_g \leq q_g^{\max} \tag{C.1c}$$

$$v^{\min} \leq |v| \leq v^{\max} \tag{C.1d}$$

$$(\angle v)_{\mathcal{R}} = \phi_{\text{slack}} \tag{C.1e}$$

$$(p_g)_{\mathcal{D}} = (q_g)_{\mathcal{D}} = 0 \tag{C.1f}$$

$$p_g - p_d - \text{diag}(v_r)(W_r v_r - W_i v_i) - \text{diag}(v_i)(W_i v_r + W_r v_i) = 0 \tag{C.1g}$$

$$q_g - q_d + \text{diag}(v_r)(W_i v_r + W_r v_i) - \text{diag}(v_i)(W_r v_r - W_i v_i) = 0 \tag{C.1h}$$

$$\text{where } v_r = |v|\cos(\angle v) \text{ and } v_i = |v|\sin(\angle v)$$

(While we write the problem in this form to minimize notation, in practice, some of the constraints in the above problem – e.g., (C.1b), (C.1c), and (C.1f) – can be condensed.)

## C.2 OVERALL APPROACH

As pointed out in Zamzam & Baker (2019), given $p_d, q_d, (p_g)_{\mathcal{G}}$ and $|v|_{\mathcal{B}\setminus\mathcal{D}}$, the remaining variables $(p_g)_{\mathcal{R}}, (q_g)_{\mathcal{B}\setminus\mathcal{D}}, |v|_{\mathcal{D}}$, and $(\angle v)_{\mathcal{B}\setminus\mathcal{R}}$ can be recovered via the power flow equations (C.1g)–(C.1h).

As such, our implementation of DC3 may be outlined as follows.

- **Input:** $x = \begin{bmatrix} p_d^T & q_d^T \end{bmatrix}^T$. (The constant $\angle v_{\mathcal{R}}$ is fixed across problem instances.)

- **Neural network:** Output $\alpha \in [0,1]^{|\mathcal{G}|}$ and $\beta \in [0,1]^{|\mathcal{B}|-|\mathcal{D}|}$ (applying a sigmoid to the final layer of the network), and compute:

$$(p_g)_{\mathcal{G}} = \alpha(p_g^{\min})_{\mathcal{G}} + (1-\alpha)(p_g^{\max})_{\mathcal{G}},$$

$$|v|_{\mathcal{B}\backslash\mathcal{D}} = \beta v_{\mathcal{B}\backslash\mathcal{D}}^{\min} + (1-\beta)v_{\mathcal{B}\backslash\mathcal{D}}^{\max}.$$

(In other words, output a partial set of variables, and enforce box constraints on this set of variables via sigmoids in the neural network.)

- **Completion procedure:** Given $z = \begin{bmatrix} (p_g)_{\mathcal{G}}^T & |v|_{\mathcal{B}\backslash\mathcal{D}}^T \end{bmatrix}^T$, solve Equations (C.1g)–(C.1h) for the remaining quantities $(p_g)_{\mathcal{R}}$, $(q_g)_{\mathcal{B}\backslash\mathcal{D}}$, $|v|_{\mathcal{D}}$, and $(\angle v)_{\mathcal{B}\backslash\mathcal{R}}$ as described below. Output all decision variables $y = \begin{bmatrix} (p_g)_{\mathcal{G}}^T & (q_g)_{\mathcal{G}}^T & |v|^T & (\angle v)^T \end{bmatrix}^T$.

- **Correction procedure:** Correct $y$ using the gradient-based feasibility correction procedure described in §3.2.

- **Backpropagate:** Compute the loss and backpropagate as described below to update neural network parameters. Repeat until convergence.

We now describe the forward and backward passes through the completion procedure, where we have introduced several "tricks" that significantly reduce the computational cost, including some that are specific to the ACOPF setting.

For ease of notation throughout, we will let $J \in \mathbb{R}^{1+2|\mathcal{B}|}$ denote the Jacobian of the equality constraints (C.1e)–(C.1h) with respect to the complete vector $y$ of decision variables. This matrix, which we pre-compute, will be useful throughout our computations.

## C.3 SOLVING THE COMPLETION

The neural network outputs $(p_g)_{\mathcal{G}}$ and $|v|_{\mathcal{B}\backslash\mathcal{D}}$ are the inputs to the completion procedure. Theoretically, we could use Newton's method to solve for all additional variables from the equality constraints. However, in practice we find that solving for all variables using Newton's method is sometimes unstable. Fortunately, we can divide the completion procedure into two substeps, where Step 1 invokes Newton's method to identify some of the variables, while Step 2 solves in closed form for the others. This greatly improves the stability of the completion procedure.

**Step 1.** Compute $|v|_{\mathcal{D}}$, $(\angle v)_{\mathcal{B}\backslash\mathcal{R}}$ via Newton's method using real power flow constraints (C.1g) at buses $\mathcal{B} \setminus \mathcal{R}$, and reactive power constraints (C.1h) at buses $\mathcal{D}$ (note that the number of equations matches the number of variables being identified). We initialize Newton's method by fixing variables determined by the neural network and those already determined in Step 1 of the completion process, and initializing $|v|_{\mathcal{D}}$, $(\angle v)_{\mathcal{B}\backslash\mathcal{R}}$ at generic initial values (these are provided in the ACOPF task setup and are typically used to initialize state-of-the-art solvers). Note that the remaining variables, those determined in Step 2 of the completion process, do not actually appear in the relevant equality constraints and therefore do not need to be set.

Let $J_{\text{Step 1}}$ denote the submatrix of $J$ corresponding to the equality constraints (C.1g)$_{\mathcal{B}\backslash\mathcal{R}}$ and (C.1h)$_{\mathcal{D}}$ and the voltage variables $|v|_{\mathcal{D}}$ and $(\angle v)_{\mathcal{B}\backslash\mathcal{R}}$ that we are solving in this step. At the $t$th step of Newton's method, let $h_{\text{Step 1}}(t) \in \mathbb{R}^{|\mathcal{B}|-|\mathcal{R}|+|\mathcal{D}|}$ denote the vector of values on the left-hand side of the relevant equality constraints (which we wish to equal identically zero), evaluated at the current setting of all problem variables. Our Newton's method updates are then

$$\begin{bmatrix} |v|_{\mathcal{D}} \\ (\angle v)_{\mathcal{B}\backslash\mathcal{R}} \end{bmatrix}_{t+1} = \begin{bmatrix} |v|_{\mathcal{D}} \\ (\angle v)_{\mathcal{B}\backslash\mathcal{R}} \end{bmatrix}_t - J_{\text{Step 1}}^{-1} h_{\text{Step 1}}(t). \tag{C.2}$$

**Step 2.** Compute the remaining variables $(p_g)_{\mathcal{R}}$ and $(q_g)_{\mathcal{B}\backslash\mathcal{D}}$ via the remaining equality constraints – that is, the real power flow equations (C.1g) at buses $\mathcal{R}$ and the reactive power flow questions (C.1h) at buses in $\mathcal{B}\backslash\mathcal{D}$.

Steps 1 and 2 together allow us to complete all the decision variables.

### C.4 BACKPROPAGATING THROUGH THE COMPLETION

Let $z$ denote the input to the completion procedure and let $z_1$ and $z_2$ denote the variables respectively derived during Steps 1 and 2 of completion. That is:

$$z \equiv \begin{bmatrix} (p_g)_{\mathcal{G}} \\ |v|_{\mathcal{B}\backslash\mathcal{D}} \end{bmatrix}, \qquad z_1 \equiv \begin{bmatrix} |v|_{\mathcal{D}} \\ (\angle v)_{\mathcal{B}\backslash\mathcal{R}} \end{bmatrix}, \qquad z_2 \equiv \begin{bmatrix} (p_g)_{\mathcal{R}} \\ (q_g)_{\mathcal{B}\backslash\mathcal{D}} \end{bmatrix}.$$

Then, we wish to backpropagate gradients of an arbitrary loss function $\ell(x, z, z_1, z_2)$, both in order to train our neural network and to perform our gradient-based correction procedure. That is, we must compute the total derivative $\mathrm{d}\ell/\mathrm{d}z$ given the partial derivatives $\partial\ell/\partial z, \partial\ell/\partial z_1$, and $\partial\ell/\partial z_2$.

Applying the chain rule through the two steps of the completion procedure, we have:

$$\frac{\mathrm{d}\ell}{\mathrm{d}z} = \frac{\partial\ell}{\partial z} + \frac{\partial\ell}{\partial z_1}\frac{\partial z_1}{\partial z} + \frac{\partial\ell}{\partial z_2}\frac{\partial z_2}{\partial z} + \frac{\partial\ell}{\partial z_2}\frac{\partial z_2}{\partial z_1}\frac{\partial z_1}{\partial z}. \tag{C.3}$$

We now consider each of these terms in this equality, except for $\partial\ell/\partial z, \partial\ell/\partial z_1, \partial\ell/\partial z_2$, which we may assume have already been computed.

**Step 2.** Let $J_{\text{Step 2}}$ denote the submatrix of $J$ corresponding to the partial derivatives of the equality constraints used in Step 2 with respect to the voltage variables $|v|^T, (\angle v)^T$. Then, we have:

$$\frac{\partial \left[ (p_g)_{\mathcal{R}}^T \quad (q_g)_{\mathcal{B}\backslash\mathcal{D}}^T \right]^T}{\partial \left[ |v|^T \quad (\angle v)^T \right]^T} = -J_{\text{Step 2}}. \tag{C.4}$$

As $\frac{\partial (q_g)_{\mathcal{B}\backslash\mathcal{D}}}{\partial (p_g)_{\mathcal{G}}} = 0$, this gives us both the terms $\partial z_2/\partial z_1, \partial z_2/\partial z$ in (C.3).

**Step 1.** Consider the total differential through the equality constraints (C.1g)$_{\mathcal{B}\backslash\mathcal{R}}$ and (C.1h)$_{\mathcal{D}}$, where all terms besides $W_r$ and $W_i$ are viewed as parameters through which to differentiate. We rearrange this total differential to put differentials of input quantities to Step 1 on one side and differentials of outputs from Step 1 on the other side:

$$J_{\text{Step 1}} \begin{bmatrix} \mathrm{d}(|v|)_{\mathcal{D}} \\ \mathrm{d}(\angle v)_{\mathcal{B}\backslash\mathcal{R}} \end{bmatrix} = \begin{bmatrix} -\mathrm{d}(p_g)_{\mathcal{B}\backslash\mathcal{R}} + \mathrm{d}(p_d)_{\mathcal{B}\backslash\mathcal{R}} \\ \mathrm{d}(q_d)_{\mathcal{D}} \end{bmatrix} - J_{\text{Step 1b}} \begin{bmatrix} \mathrm{d}|v|_{\mathcal{B}\backslash\mathcal{D}} \\ \mathrm{d}(\angle v)_{\mathcal{R}} \end{bmatrix}, \tag{C.5}$$

where $J_{\text{Step 1}}$ is as in the forward pass of Step 1, and $J_{\text{Step 1b}}$ denotes the submatrix of $J$ corresponding to the equality constraints (C.1g)$_{\mathcal{B}\backslash\mathcal{R}}$ and (C.1h)$_{\mathcal{D}}$ and the voltage variables $(|v|)_{\mathcal{B}\backslash\mathcal{D}}, (\angle v)_{\mathcal{R}}$.

While we could compute $\begin{bmatrix} \mathrm{d}|v|_{\mathcal{D}} \\ \mathrm{d}(\angle v)_{\mathcal{B}\backslash\mathcal{R}} \end{bmatrix}$ explicitly, we can improve efficiency by not computing and storing a large intermediate Jacobian explicitly. Instead, we can directly compute what we need, which is the product of this Jacobian with matrices of smaller dimensions. Define

$$K \equiv \left( \frac{\partial\ell}{\partial z_1} + \frac{\partial\ell}{\partial z_2}\frac{\partial z_2}{\partial z_1} \right) J_{\text{Step 1}}^{-1},$$

using our computation (C.4) above to evaluate $\partial z_2/\partial z_1$. Note furthermore that $J_{\text{Step 1}}^{-1}$ was already computed in the forward pass, meaning that we do not need to perform an additional matrix inversion.

Now, we can use (C.5) to derive:

$$\left(\frac{\partial \ell}{\partial z_1} + \frac{\partial \ell}{\partial z_2}\frac{\partial z_2}{\partial z_1}\right)\mathrm{d}z_1 = K\left(\begin{bmatrix} -\mathrm{d}(p_g)_{\mathcal{B}\setminus\mathcal{R}} + \mathrm{d}(p_d)_{\mathcal{B}\setminus\mathcal{R}} \\ \mathrm{d}(q_d)_{\mathcal{D}} \end{bmatrix} - J_{\text{Step 1b}}\begin{bmatrix} \mathrm{d}(|v|)_{\mathcal{B}\setminus\mathcal{D}} \\ \mathrm{d}(\angle v)_{\mathcal{R}} \end{bmatrix}\right),$$

which gives us

$$\frac{\partial \ell}{\partial z_1}\frac{\partial z_1}{\partial z} + \frac{\partial \ell}{\partial z_2}\frac{\partial z_2}{\partial z_1}\frac{\partial z_1}{\partial z}.$$

**Putting it all together.** Combining our logic described above for Steps 2 and 1, we can recover all terms in (C.3) and therefore identify $\mathrm{d}\ell/\mathrm{d}z$.

