# OpenReview forum: "DC3: A learning method for optimization with hard constraints"
_ICLR.cc/2021/Conference — ICLR 2021 Poster_

### Official Review · AnonReviewer3 · 2020-10-22
**Not very pretty, but (to my knowledge) first method to deal with arbitrary nonlinear constraints**

**Rating:** 7
**Confidence:** 4

**Review:**

There has been an increase of works using deep neural networks to heuristically predict solutions to constrained optimization problems. However, these methods cannot generalize to arbitrary constraints.  In this paper, the authors propose a method to build neural networks that output vectors that satisfy hard equality and inequality constraints. They do this by first having the network predict the underdetermined part of the system defined by the equalities, then doing a series of gradient steps to project the solution onto the space delineated by the inequalities. They evaluate on synthetic quadratic programs and problems derived from a AC power flow application.

Prior to this work I could see two ways of doing what they propose.

A) If the constraints are linear, then Frerix et al. (2020) propose to compute the Minkowski-Weyl decomposition of the polyhedron of constraints, and to have the neural net output the coefficients in the resulting basis. Computing the MW decomposition is very expensive (both in theory and in practice), but once it is computed, enforcing the constraints is automatic.

B) Alternatively, if the constraints are (disciplined) convex, one could use a differentiable convex layer as last layer, e.g. using the method of Agrawal et al. (2019) with a somewhat arbitrary choice of objective function. For example, it could be a quadratic program layer (as in the preceding work of Amos and Kolter (2017), where a neural net with a final QP layer is used to output soft solutions to Sudokus, which the authors do cite). In fact when constraints are convex something very similar to this paper can be done with this approach, where a neural net would predict a solution, and a final QP layer would find the point that satisfies the constraints that is the closest in L2 distance with the initial neural net guess.

The current paper discusses briefly works related to method B in the "Implicit layers" paragraph of Section 2, without discussing how it can accomplish what they aim to do, nor compare against it. It is a direct competitor when constraints are convex. As for method A, it is neither cited nor compared against. Against, this is a direct competitor when constraints are linear.

So in what way does the current approach compare to these previous ones? The main advantage I see is that the method can be applied to nonconvex problems. Of course, this doesn't mean it will necessarily do well - the Newton's method step for the equality constraints or the first-order projection for the inequality constraints might not converge. There is no miracle and probably this method won't be workable on nonlinear constraints that are too wild. But at least, it seems it does converge for the nonconvex "ACOPF" problem of the experiments, so there seems to be some use-case. I think this advantage (the method can be applied to arbitrary nonlinear constraints) should be emphasized more, and a discussion of the probable limitations should be included as well. In particular, when more assumptions are put on the constraints (e.g. linearity), there are probably better options than first-order methods.

Regarding the timings in the experiments: all the machine learning methods use a GPU, so it would be interesting to compare against a baseline solver that can use a GPU when possible as well. For example, for the QP, the qpth library of Amos and Kolter (2017) has the entire QP solver implemented on CUDA - this might make a difference in speed.

Overall, my opinion is that the paper does nothing groundbreaking, but has the quality of being the first method (as far as I know) to propose enforcing hard equality-inequality constraints on the output of a neural net, which is an obstacle right now in extending recent work on learning heuristics for optimization problems. The literature review definitely has to be expanded however, to explain that when constraints are linear and convex respectively, the works of Frerix et al. (2020) [method A] and Agrawal et al. (2019) [method B] are competitors. Moreover, more discussion on the advantages and inconvenients of the approach is necessary (i.e. that the method is very general, but probably inefficient on more structured problems, and yet probably can't work on constraints that are too wild). But otherwise, the core concept of the paper sounds sound to me, and the experiments, although minimal, are I think correctly made (several seeds, etc.) I also appreciate that code was provided. So I tend towards weak acceptance of the paper.

In case this paper is rejected, my recommendation to the authors would be strengthen the experimental section. First, since comparisons on QPs are presented, I would have liked to see comparisons against these methods. I think comparisons on another nonconvex problem would be a must as well. Also comparisons in a "predict-then-optimize" setup (e.g. see Elmachtoub and Grigas 2017, Donti et al. 2017, Wilder et al. 2019, Vlastelica et al. 2019) would be very welcome, since this is a setup where neural networks have tremendous potential. This would make for a much stronger paper.


References

[1] Frerix, T., Nießner, M., & Cremers, D. (2020). Homogeneous linear inequality constraints for neural network activations. In Proceedings of the IEEE/CVF Conference on Computer Vision and Pattern Recognition Workshops (pp. 748-749).

[2] Agrawal, A., Amos, B., Barratt, S., Boyd, S., Diamond, S., & Kolter, J. Z. (2019). Differentiable convex optimization layers. In Advances in neural information processing systems (pp. 9562-9574).

[3] Amos, B., & Kolter, J. Z. (2017). Optnet: Differentiable optimization as a layer in neural networks. arXiv preprint arXiv:1703.00443.

[4] Elmachtoub, A. N., & Grigas, P. (2017). Smart" predict, then optimize". arXiv preprint arXiv:1710.08005.

[5] Donti, P., Amos, B., & Kolter, J. Z. (2017). Task-based end-to-end model learning in stochastic optimization. In Advances in Neural Information Processing Systems (pp. 5484-5494).

[6] Wilder, B., Ewing, E., Dilkina, B., & Tambe, M. (2019). End to end learning and optimization on graphs. In Advances in Neural Information Processing Systems (pp. 4672-4683).

[7] Vlastelica, M., Paulus, A., Musil, V., Martius, G., & Rolínek, M. (2019). Differentiation of blackbox combinatorial solvers. arXiv preprint arXiv:1912.02175.

---

Edit after rebuttal: I am satisfied with the changes to the paper and increase my score to an accept.

---

> ### Author Response · Authors · 2020-11-21
> **Response to R3**
>
> Thank you for your thoughtful consideration of our paper. Before responding to your individual points, we would like to confirm that while we do test on “easy” QP instances, the problem of real interest to us is the non-convex, nonlinearly-constrained problem of ACOPF (which was indeed our main motivation for writing this paper). We have added some language throughout the paper in an attempt to make this clearer, but (as you suggest) plan to make this more explicit in the final version of the paper.
>
> To address your specific points:
>
> > The current paper discusses briefly works related to method B in the "Implicit layers" paragraph of Section 2, without discussing how it can accomplish what they aim to do, nor compare against it. It is a direct competitor when constraints are convex.
>
> Thank you for this note; we have now included additional discussion on this point in the “implicit layers” section of the related work. Interestingly, this was indeed our first instinct when we started to approach this overall problem: That we could simply use differentiable optimization layers to project onto constraint sets in convex cases. However, we quickly abandoned this approach for two reasons:
> * For one, this is a somewhat computationally expensive proposition. For instance, when we applied qpth (Amos and Kolter (2017)) directly to the QP settings we consider (described more below), this was much slower than using the DC3 approach. (While we realize this is not the same thing as using qpth for constraint projections, we bring this up as a proxy for the overall computational expense.) Nonetheless, we will plan on including a direct comparison to projection via qpth/convex optimization layers in a future version of our paper.
> * As you point out, this approach is not available in settings with non-convex constraint sets, whereas the settings of greatest interest to us (e.g. ACOPF) have nonlinear, non-convex constraints.
>
>
> > As for method A, it is neither cited nor compared against. Against, this is a direct competitor when constraints are linear.
>
> We have now included a citation to Frerix et al. (2020) in the section on “constraints in neural networks” within the related work. Based on our understanding, however, this framework in its current form could not necessarily be directly applied even in the QP contexts we consider. In particular, Frerix et al. (2020) enforce linear inequality constraints of the form $Ax \leq 0$ in deep learning model activations. While their conceptual framework could in theory be extended to more general sets of linear constraints of the form $Ax \leq b$ (which would cover our QP setup), the authors mention that this setting is not directly amenable to the efficient constraint representation they employ in the homogenous case, and that in practice experiments in this more general setting suffered from slow convergence due to vanishing gradients (see Section 3.5 of Frerix et al. (2020)). As such, while we fully intend to compare against this MW decomposition framework in future versions of our submission, this may require additional steps beyond simply re-implementing the method proposed in Frerix et al. (2020).

---

> > ### Author Response · Authors · 2020-11-21
> > **Response to R3 (ctd.)**
> >
> > > I think this advantage (the method can be applied to arbitrary nonlinear constraints) should be emphasized more, and a discussion of the probable limitations should be included as well. In particular, when more assumptions are put on the constraints (e.g. linearity), there are probably better options than first-order methods.
> >
> > > Moreover, more discussion on the advantages and inconvenients of the approach is necessary (i.e. that the method is very general, but probably inefficient on more structured problems, and yet probably can't work on constraints that are too wild)
> >
> > Agreed. We have added some text throughout the paper to emphasize that our method can be applied to nonlinear constraints that play nicely with gradient descent, and that our method is not likely the “best” choice in the case of linear constraints.
> >
> > > Regarding the timings in the experiments: all the machine learning methods use a GPU, so it would be interesting to compare against a baseline solver that can use a GPU when possible as well. For example, for the QP, the qpth library of Amos and Kolter (2017) has the entire QP solver implemented on CUDA - this might make a difference in speed.
> >
> > This is a good point. We have now added a comparison in the paper against qpth. We find that qpth is about 40x slower than our approach (though of course it solves the QP instances “exactly,” whereas DC3 does not achieve optimal objective values in the QP setting).
> >
> > > In case this paper is rejected, my recommendation to the authors would be strengthen the experimental section. First, since comparisons on QPs are presented, I would have liked to see comparisons against these methods. I think comparisons on another nonconvex problem would be a must as well. Also comparisons in a "predict-then-optimize" setup (e.g. see Elmachtoub and Grigas 2017, Donti et al. 2017, Wilder et al. 2019, Vlastelica et al. 2019) would be very welcome, since this is a setup where neural networks have tremendous potential. This would make for a much stronger paper.
> >
> > Thank you for these suggestions!

---

### Official Review · AnonReviewer4 · 2020-10-28
**Strong paper but the central claim is compromised**

**Rating:** 8
**Confidence:** 4

**Review:**

This paper proposes a method to strictly enforce hard constraints during a neural network, without compromising differentiability. The method has two stages 1) From a smaller set of predicted variables, compute the remaining ones so that equality constraints are satisfied; 2) Take a few gradient steps (w.r.t soft constraint) in case inequality constraints are violated. They perform experiments on synthetic and also somewhat applied instances of quadratic programs. The results look very promising.

While most of this review will focus on the negative aspect, I want to begin by stating that I like the paper very much.

I think it solves an important problem (from a fundamental standpoint but with potentially large impact), it solves it with a method that is non-trivial, yet quite simple (both of which are positives) and shows performance clearly better than that of naive -- but popular -- baselines. The writeup is very nice and the experiments are sufficient.

Having said all of that, I cannot recommend acceptance of the paper in its current stage. In several places, the paper claims to have orders-of-magnitude faster runtimes than standard solvers. It is even the only claim of the paper appearing in boldface. Since the time measurement methodology is not described in the paper, I went into the code attached to the submission. Below is a list of reasons why I believe the current methodology is flawed and does not permit making strong claims from the paper.

- CVXPy was never developed to be a fast solver. I checked this with the CVXPy authors and often times the solver is not even the computational bottleneck.
- Even if other bottlenecks are factored out of CVXPy, there is an option to use an almost-SOTA QP solver on the backend with prob.solve(solver=cp.OSQP), which the authors **do not use**. For these two reasons, it would probably be better to compare against the freely available OSQP directly, without incorporating CVXPy -- (I feel, CVXPy, in general, shouldn't be promoted as a baseline for runtimes)
- I didn't go through all the code but the time measurements I found were around large blocks of code, not the pure runtime of the solvers. This leaves a lot of room for inaccuracies.
- The baselines optimizers do not enjoy any parallelization over the instances. On the other hand, the authors write (in a footnote on page 11) that their method was timed with *full parallelization* (all instances in one batch). **This is massively unfair to the baseline** and **may fully explain the orders of magnitude difference in runtimes**.
- Runtime of its own is a bit of a compromised quantity due to hardware (cpu/gpu) differences, maybe FLOPS would be fairer (but I understand it might not be feasible)

There are two ways forward:

a) The authors remove/strongly-tone-down all claims about runtimes. In this case, I am ready to dramatically improve my evaluation as I believe the paper is safely above the acceptance threshold even without them.

b) The authors insist on claims about runtime. If this is the case, the authors need to address the points made above, update the results, and add a section that demonstrates the soundness of the methodology (possibly into supplementary). In such a case, I am happy to reevaluate but I will still insist on a proper justification of the claims made in boldface.

---

> ### Author Response · Authors · 2020-11-21
> **Response to R4**
>
> Thank you so much for your thoughtful review. First of all, we would like to sincerely apologize for the way in which the runtimes were presented in the original version of our paper, and thank you for pointing this out. In particular, we had meant to more explicitly highlight the point regarding parallelization within the main body of the paper, but this accidentally fell through the cracks as we moved content between the main paper and the appendix. We have now revised the paper to make this point more explicit in the main body of the text, and have also incorporated the rest of your suggestions in this revised version.
>
> More explicitly:
>
> > it would probably be better to compare against the freely available OSQP directly, without incorporating CVXPy
>
> Thank you for this suggestion. In the revised version, we have run timing results with OSQP instead of with CVXPY (and also tightened the timing blocks to surround only the solver, as you mention elsewhere in your review). In this particular case, the difference did not end up being that large — previously, we timed 43.28 $\pm$ 0.81 for CVXPY, whereas we now have 41.68 $\pm$ 4.80 for OSQP — but you are right that OSQP is indeed the correct comparison.
>
> > the time measurements I found were around large blocks of code, not the pure runtime of the solvers. This leaves a lot of room for inaccuracies.
>
> This is a good point — we had previously timed the function calls to our `opt_solve` function for the baseline optimizer, but you are correct that we should have more tightly timed the call to the actual solver within that function (rather than also timing the setup code). We have now modified the timing code for the baseline optimizers within both the QP and ACOPF settings. As described above, for the QP setting, this did not end up making much of a difference. For the ACOPF setting, this changed the baseline timing result from 182.71 $\pm$ 19.29 to 103.35  $\pm$ 4.48.
>
> > The baselines optimizers do not enjoy any parallelization over the instances. On the other hand, the authors write (in a footnote on page 11) that their method was timed with full parallelization (all instances in one batch).
>
> Thank you for taking the time to look into our code and appendix! Again, we sincerely apologize for not making this clearer in the main text; this omission was inadvertent. We have now updated the captions of all tables and the main text in Section 4 to make it clearer that there is a difference in how we are employing parallelization between the deep learning-based methods vs. the baseline optimizers.
>
> To explain our rationale for why we are reporting timing results in this way: We agree that in theory, if the OSQP and PYPOWER solvers (in the QP and ACOPF settings, respectively) were run with full parallelization, then they would actually beat our DC3 method in terms of wall clock time (though of course, as you mention, this comparison is not entirely apples-to-apples due to the difference in CPU vs. GPU). However, in many practical instances, it is not easy for practitioners to take advantage of parallelization for traditional optimizers as easily as they can for methods run within deep learning frameworks. In particular, per the timing results in the revised version of the paper, practitioners would need to use ~350 threads of parallelization (in the QP case) and ~45 threads of parallelization (in the ACOPF case) to match the wall clock speed of our solver. Depending on the hardware available, this level of parallelization may not be readily achievable.
>
> Given the nuance here, we have updated the discussion in Section 4 of our paper to ensure that we (a) mention the caveat about parallel vs. sequential runs, and (b) hint at how “easily” these different methods can be parallelized in every instance where we mention speedup factors.
>
>  We have also added in our revision a comparison to qpth (the QP solver developed for OptNet [1]), which can run with full parallelization on a GPU. We find that our DC3 method achieves a 40x speedup over qpth (though, by construction, qpth achieves exactly optimal objective values, whereas DC3 does not in the QP setting).
>
> [1] Amos, B., & Kolter, J. Z. (2017). Optnet: Differentiable optimization as a layer in neural networks. arXiv preprint arXiv:1703.00443.
>
> > Runtime of its own is a bit of a compromised quantity due to hardware (cpu/gpu) differences, maybe FLOPS would be fairer (but I understand it might not be feasible)
>
> Agreed. Given these factors and many other sources of uncertainty that can additionally arise (also because we did not code our DC3 implementation in a way that was optimized for speed), we have also updated our description of the timing comparisons in Section 4 to emphasize that they are relatively approximate (see the last paragraph before Section 4.1).

---

> > ### Author Response · Authors · 2020-11-21
> > **Response to R4 (ctd.)**
> >
> > > There are two ways forward:
> > a) The authors remove/strongly-tone-down all claims about runtimes.
> >
> > As hinted at above, we have gone through the paper and toned down/added more discussion to the claims about runtimes. The major changes here include:
> > * Modifying the key contributions highlighted in the introduction, in particular to remove the point about “significant speedup.”
> > * In the experiments section, in every table, we have now added additional description in the captions to point out the sequential vs. parallel issue, and also put a superscript next to the timing numbers reported for optimizers run in parallel to emphasize that these numbers are different in character than the other numbers reported.
> > * In every place where we report a speedup factor, we have now added a discussion of the sequential vs. parallel issue, as well as some discussion of the nuance about the ease of parallelizability of different methods.
> > * Throughout the writing, we have removed statements about achieving significant speedup over the baseline optimizers.
> >
> > Thank you again for your comments, and for your thoroughness in reviewing our paper. We greatly appreciate the feedback you have given, and would welcome any additional suggestions for improving the paper.

---

### Official Review · AnonReviewer1 · 2020-10-29
**The paper proposes a learning based framework for the solving optimization problems with fixed equality and inequality constraints in the formulation. Experiments on quadratic programming instantiated with synthetic data and on AC optimal power flow problems are presented for the proposed framework.**

**Rating:** 4
**Confidence:** 5

**Review:**

Strength:
+ The paper proposes a general framework to deal with constraints in optimization problems using neural networks. In my opinion this is an important problem since there exists no standard method in many existing deep neural network frameworks to deal with constraints, which are also inapplicable even if the constraints are only slightly nontrivial. The paper proposes to deal with equality and inequality constraints differently which may be often easier in large scale settings.
+ Comparison with CVXPY indicates that the algorithm can indeed be practically useful, and applicable for a broad spectrum of problems (this is not discussed in the paper).

Weakness:
- While the related work from the recent years has been discussed to some extent, the paper fails to show how the technique proposed is better than them. The main idea of the paper has a rich history in optimization literature and is referred to as "elimination" of constraints. See Chapter 15 in Numerical Optimization by Nocedal & Wright (2006), and references within for more details. From my understanding, the discussion in Amos & Kolter (and their implementation) makes it clear that a reduced/partial Cholesky factorization is sufficient. This is equivalent to the completion procedure suggested here. Moreover, the correction procedure simply reduces a penalized form of constraints, and is not a projection operation. To this end, the paper does not have any discussion regarding the convergence aspects of the framework which is a crucial subject for this paper.
- For a generic optimization framework (proposed), the paper only provides experiments on well studied optimization problems - all the experiments provided in the paper are with quadratic programming problems which are known to be easy both in theory and practice. It will be interesting to see how the framework performs in different problems. Also, it is not clear whether the experimental benefits will carry to other networks that are used in practice. Indeed, running experiments on all known network architectures is feasible but showing experiments on a few more relevant architectures can add a lot of value to the paper and readers.
After response: While some of my concerns were cleared after the response, the experimental evaluations presented do not sufficiently support the claim that the method can handle nonlinear constraints.

---

> ### Author Response · Authors · 2020-11-21
> **Response to R1**
>
> Thank you for the careful reading and these very helpful comments. Before addressing your specific points, we would like to clarify that while we test on simulated (convex) QP instances to illustrate our method, the more realistic experimental test lies is in addressing the problem of AC optimal power flow, which is a non-convex optimization problem (with nonlinear, non-convex constraints) that is challenging to solve in practice. As such, we see one of the key benefits of our method as its ability to be applied to non-convex optimization problems with arbitrary constraints, where solvers do not necessarily come with global optimality guarantees (and cannot always exploit tricks such as Cholesky decomposition). We fully agree that we did not make this point sufficiently clear, and also that we should have included more discussion of prior work; we have updated the paper in this regard.
>
> In response to the specific points you raise:
>
> > The main idea of the paper has a rich history in optimization literature and is referred to as "elimination" of constraints.
>
> We have strengthened the discussion of related work as suggested. We certainly do not mean to have implied that elimination of variables itself is an innovation - rather, the innovation is the application of variable elimination within a general-purpose deep learning framework for optimization.
>
> > From my understanding, the discussion in Amos & Kolter (and their implementation) makes it clear that a reduced/partial Cholesky factorization is sufficient.
>
> We have revised the paper to include a comparison to OptNet (Amos & Kolter) in the case of the QP task (the results indicate that our method is significantly faster, though does not solve the problem exactly). To clarify, OptNet leverages a full Cholesky decomposition on a symmetrized KKT matrix. However, since the ACOPF setting we apply is non-convex (specifically, the constraints are nonlinear and non-convex), we are not able to directly employ OptNet/Cholesky decomposition for comparison.
>
> > the paper does not have any discussion regarding the convergence aspects of the framework which is a crucial subject for this paper.
>
> We have strengthened our discussion of convergence for the correction procedure. While gradient-based methods are not in general guaranteed to converge, they do converge in practice if initialized close to the global optimum. In our case, the output of the completion procedure is already close to optimal at test time (since the network is trained using a soft loss), explaining the effectiveness of gradient-based correction.
>
> > the paper only provides experiments on well studied optimization problems - all the experiments provided in the paper are with quadratic programming problems which are known to be easy both in theory and practice.
>
> AC optimal power flow is actually a hard, non-convex optimization problem (which we make clearer in our revision). There is an urgent need in practice for approximate optimization algorithms on this task, as current algorithms used in operating the electrical grid do not scale well. By comparison, the convex QP task we consider is indeed very simple and was intended as an illustration.
>
> > Also, it is not clear whether the experimental benefits will carry to other networks that are used in practice. Indeed, running experiments on all known network architectures is feasible but showing experiments on a few more relevant architectures can add a lot of value to the paper and readers.
>
> The general DC3 framework is agnostic as to the network architecture, something we make clearer in our revision. We used a simple network (consistent across all methods tested) to better illustrate the effect of our methodology. Can you clarify if there are other architectures you would recommend we try for the QP or ACOPF tasks, and what insights could be gained from them?

---

### Official Review · AnonReviewer2 · 2020-10-29
**Interesting, well-motivated method**

**Rating:** 6
**Confidence:** 4

**Review:**

### quality
Good

### clarity
Good

### originality
Carefully combines a number of non-trivial methods from recent related work. Clever new idea for how to guarantee constraint satisfaction of the output of the network.

### significance
Concrete contribution to the field of learnable 'optimizers.' The applications are well-motivated and it appears that deploying the method for the ACOPF could have immediate impact.

### Pros
Well executed + motivated. Careful ablation analysis.

### Cons
The results may be more impactful and interesting in an operations research venue. The gap between the learned optimizer and the exact QP solver is non-trivial for some problems. This may be a deal-breaker for deployment.

## Comments
### Satisfying inequality constraints =

It was unclear to me why the gradient descent procedure is guaranteed to converge. Is this assuming that the constraint set is convex?

I was confused as to why the equality and inequality constraints are treated so differently. Why have a black box solver with implicit function theorem differentiation for equality constraints vs. back-propping through unrolled gradient descent for the inequality constraints?


FYI, in principle, the optimization in 3.2 doesn't need to be in z space. It could be in hidden space of the neural network that outputs z. In future work this might allow local search to move better when seeking to enforce the inequality constraints.

### Benchmarking
4.1: Does making Q diagonal change the difficulty of the optimization problem? Is this a realistic setting?


### The 'Optimizer' Baseline=
Table 2: The gap between the exact QP optimizer and DC3 is non-trivial, with a scale that is often bigger than the gap between DC3 and some of the other deep net baselines. The claim "we find that DC3 achieves comparable objective" may be over-stated. Perhaps you should do a paired analysis across optimization problems. When comparing method X to method Y, on how what fraction of problems does X outperform Y?

Can the 'Optimizer' baseline be warm-started? If so, how fast would it be to warm start using DC3. This might help close the speed gap between 'Optimizer' and DC3 while achieving higher quality solutions.

## Minor Comments

I initially found (1) confusing because I thought that part of the learning problem was to learn the dependence of the optimization problem on x. This occurs, for example, in the field of structured prediction, where we have a labeled dataset of (x, y) pairs. You should update to emphasize that this is not the case.

In algorithm 1, line 10, aren't you doing gradient descent, not stochastic gradient descent? It should be GD.

In Table 3, DC3, \leq train matches DC3. Why do you expect this is true?


Page 5: I would have preferred to see the discussion of the very relevant related work in the main text and not relegated to an appendix.

---

> ### Author Response · Authors · 2020-11-21
> **Response to R2**
>
> Thank you for the careful reading and these very helpful comments. In response to the points you raise:
>
> > The results may be more impactful and interesting in an operations research venue.
>
> While we hope and appreciate that our contribution may also be of interest to operations researchers, we believe that our contribution is also of great importance to the deep learning community, particularly due to the difficulty that deep learning methods often have in fully satisfying constraints. Arguably, this limitation has been a major reason that deep learning methods have not necessarily found success in many real-world scientific and engineering domains where feasibility is fundamentally important. As such, we believe that methods to enforce hard constraints are an important part of the conversation in advancing deep learning research.
>
> > It was unclear to me why the gradient descent procedure is guaranteed to converge. Is this assuming that the constraint set is convex?
>
> We have strengthened our discussion of convergence for the correction procedure. While gradient-based methods are not in general guaranteed to converge, they do converge in practice if initialized close to the global optimum. In our case, the output of the completion procedure is already close to optimal at test time (since the network is trained using a soft loss), explaining the effectiveness of gradient-based correction.
>
> > I was confused as to why the equality and inequality constraints are treated so differently.
>
> We use a different procedure for equality and inequality constraints since (a) equality constraints are much harder to satisfy purely with "soft loss" or correction procedures, (b) given m equality constraints and n total variables, we know that the manifold of solutions has dimension n-m, which means we can solve directly for points on that manifold - this is in essence what our completion procedure does, (c) by contrast, there could be an arbitrarily large number of inequality constraints, (d) we are able to use the soft loss to come close to satisfying all inequality constraints, and since we are close to feasible it is possible to fix any remaining violations using gradient-based correction.
>
> > 4.1: Does making Q diagonal change the difficulty of the optimization problem? Is this a realistic setting?
>
> The QP task was chosen as a simple, convex optimization test case (while ACOPF is a hard non-convex problem). In general, the Q matrix can be any positive semi-definite matrix, however we chose a diagonal matrix for simplicity and because it is also a common case for QPs in practice. (ACOPF actually also has a diagonal matrix in the objective, even though the task is much harder since the constraints are non-convex.)

---

> > ### Author Response · Authors · 2020-11-21
> > **Response to R2 (ctd.)**
> >
> > > The gap between the learned optimizer and the exact QP solver is non-trivial for some problems. This may be a deal-breaker for deployment.
> >
> > > The claim "we find that DC3 achieves comparable objective" may be over-stated.
> >
> > As you suggest, we have modified the "comparable objective" language to guard against over-stating the results.
> >
> > > The gap between the exact QP optimizer and DC3 is non-trivial, with a scale that is often bigger than the gap between DC3 and some of the other deep net baselines.
> >
> > The objective gap between the exact QP optimizer and non-DC3 deep learning methods is very deceptive - all the other deep learning methods fail to satisfy the constraints, which means that their objective values are essentially worthless. Indeed, some of these methods claim to have found objective values significantly higher than the true optimum, which is only possible because these "solutions" are infeasible.
> >
> > > Perhaps you should do a paired analysis across optimization problems. When comparing method X to method Y, on how what fraction of problems does X outperform Y?
> >
> > This is a great suggestion. We are currently working on this, and hope to include it in the paper during this revision period (and will plan on including it in the final version of our paper nonetheless).
> >
> > > Can the 'Optimizer' baseline be warm-started?
> >
> > We had tried a warm-starting approach, but it essentially didn't change the results at all. We would be happy to include these results if you recommend we do so.
> >
> > > In Table 3, DC3, \leq train matches DC3. Why do you expect this is true?
> >
> > The reason why DC3, \leq train matches DC3 in Table 3 is likely that in ACOPF the inequality constraints are not as hard to satisfy as the equality constraints.
> >
> > > Page 5: I would have preferred to see the discussion of the very relevant related work in the main text and not relegated to an appendix.
> >
> > Our rationale for putting this discussion in a footnote was that this discussion was very specific to ACOPF, despite the fact that our overall description of the DC3 framework is more general. However, we did not want to omit this discussion, since (as you mention) the work we discuss is very relevant in the context of ACOPF. As such, we have left this discussion in a footnote, but would definitely welcome your further thoughts if you think that moving this discussion to somewhere in the main body of the paper would be useful.
> >
> > ### Other
> >
> > We have instantiated all the other minor suggestions you make.

---

### Decision · Program_Chairs · 2021-01-07
**Final Decision**

**Decision:**

Accept (Poster)

**Comment:**

The paper proposes an approach for solving constrained optimization problems using deep learning. The key idea is to separate equality and inequality constraints and "solve" for the equality constraints separately. Empirical results are given for convex QPs and for a non-convex problem that arises in AC optimal power flow.
There was much discussion of this paper between the reviewers and the area chair. THe key question was whether the empirical evaluation is sufficient to convince that the method is more effective than existing solvers. The current experiments do not show that the method achieves better solutions than existing solvers. For the convex case this is to be expected since solvers are optimal. But in the non-convex case, it would have been nice to see that the method indeed can find better solutions.
This leaves the advantage of the method in its speedup over existing methods. However, as the authors acknowledge, it is possible that this speedup is due to better use of parallelization than the methods they compare to. It is true that deep learning is particularly easy to parallelize, but this is not impossible for other methods (e.g., for linear algebra operations etc).
Thus, taken together the empirical support for the current method is somewhat limited. The method itself does make sense, and this was indeed appreciated by the reviewers.